# Methylglyoxal Decoration of Glutenin during Heat Processing Could Alleviate the Resulting Allergic Reaction in Mice

**DOI:** 10.3390/nu12092844

**Published:** 2020-09-17

**Authors:** Yaya Wang, Xiang Li, Sihao Wu, Lu Dong, Yaozhong Hu, Junping Wang, Yan Zhang, Shuo Wang

**Affiliations:** 1Tianjin Key Laboratory of Food Science and Health, School of Medicine, Nankai University, Tianjin 300071, China; 18829349424@163.com (Y.W.); lixiang0015@mail.nankai.edu.cn (X.L.); 2120181359@mail.nankai.edu.cn (S.W.); 819063@nankai.edu.cn (L.D.); yzhu@nankai.edu.cn (Y.H.); yzhang@tust.edu.cn (Y.Z.); 2College of Food Engineering and Biotechnology, Tianjin University of Science & Technology, Tianjin 300457, China; wangjp@tust.edu.cn

**Keywords:** glutenin, methylglyoxal, allergic reaction, gut microflora, heat-processing

## Abstract

Background: It is widely believed that Maillard reactions could affect the sensitization of allergens. However, the mechanism of action of methylglyoxal (MGO) production in Maillard reactions in the sensitization variation of glutenin (a predominant allergen in wheat) during heat processing is still unclear. Methods: This research evaluated the effect of MGO on the immune response against glutenin in a mouse model. The resulting variations in conformation and corresponding digestibility of glutenin were determined. The immune response and gut microflora variation in mice were analyzed following administering of glutenin and MGO-glutenin. Results: The results of the study showed that MGO-glutenin induced a lower immune response than native glutenin. Cytokine analysis showed that MGO-glutenin regulated mouse immune response by inducing Treg differentiation. MGO decoration changed the structure and digestibility of glutenin. In addition, MGO-glutenin contributes to the maintenance of the beneficial gut microflora. Conclusion: MGO decoration of glutenin during heat processing could alleviate the resulting allergic reaction in mice. Decoration with MGO appears to contribute to the aggregation of glutenin, potentially masking surface epitopes and abating sensitization. Furthermore, Bacteroides induced regulatory T-cell (Treg) differentiation, which may contribute to inhibition of the Th2 immune response and stimulation of immune tolerance.

## 1. Introduction

As an important food resource, wheat has been processed into diverse foodstuffs to meet different tastes. While high consumption of wheat can ensure adequate energy and nutrition supply it can also increase the risk of developing allergic disease, including mild and acute reactions induced by wheat allergens. This has prompted research into studying the food allergies of wheat (FAW) toward determining appropriate interventions or for alleviation of allergic reactions. As the predominant allergen, gluten is involved in this immune disorder, especially in the form of glutenin, which is the main protein in terms of FAW [1].

Over the past few decades, the consumption of ultraprocessed foods derived from wheat has increased dramatically. On the one hand, ultraprocessing has been reported to increase the level of side products that are harmful to the host either by direct interplay or via accumulation as intermediates. On the other hand, the Maillard reaction that occurs during this process has the potential to conformationally modify allergen proteins and, as a result, alter the allergenicity of related allergens. Recently, pyrrolidine produced from heat-processed ovalbumin has been shown to have increased immunogenicity, enhancing dendritic cell uptake and IgE production [2], whereas sensitization toward vicilin was decreased after its interaction with glucose during Maillard reaction [3]. Herein, it was concluded that the complicated effects of protein glycation on sensitization are not consistent for different conditions (some increase but some decrease). Allergen processing in the digestive tract plays an important role in determining the allergenicity of food proteins. While a previous report suggested a correlation between digestive stability and allergenicity [4], further research indicated that this correlation was not rigorous [5]. In addition, the changes in allergen structure caused by heat processing are closely related to alterations in the allergenicity of food allergens [6]. An important reason for the increased stability of food allergens during heat processing is the formation of new disulfide bonds or the maintenance of inherent disulfide bonds [7]. However, the role of glycosylation of food allergens on the regulation of immunological properties remains obscure and needs to be elucidated. Gut-associated lymphoid tissue (GALT), as part of the mucosal immune system, is the main tissue responsible for the allergic reactions occurring in the digestive system, especially through oral intake. In GALT, various immune cells and cytokines are involved in the eventual immune response including the allergen-related allergic symptoms or immune tolerance. However, the mechanism by which the individuals can modulate the immune response against allergens remains unclear.

The hygiene hypothesis provides the basis for the correlation between allergic reaction and microbes and addresses environmental changes as a major factor for the development of allergies [8]. Therefore, gut microflora have been demonstrated to play a crucial role in food allergy. The evidence suggests that specific bacterial species from healthy gut microflora play an important role in regulating immune tolerance, as well as their metabolites, such as short-chain fatty acids [9]. It has been demonstrated that the imbalance of gut microflora, characterized by changes in the composition and functional imbalance of intestinal microorganisms, contributes to the development of food allergy (FA) [10]. However, the published conclusions on the features of gut microflora associated with FA still seem preliminary given the generally small number of observations [11].

The Maillard reaction is the reaction of reducing sugars and sugar degradation products with proteins. α-Dicarbonyl compounds like methylglyoxal (MGO) and glyoxal are important intermediates in the Maillard reaction. Of these, MGO has the highest reactivity. In the past, we have studied the effect of MGO on the digestibility of glutenin and its mechanism. Herein, the current project aimed to determine the effect of MGO on the allergenicity of glutenin based on the BALB/c mouse model pre-sensitized to native glutenin, heated glutenin, and MGO-glutenin in order. The changes of structure and digestibility of glutenin and gut microflora in mice were analyzed to elucidate the detailed mechanism by which the potential for allergic reaction is reduced as a result of MGO decoration.

## 2. Materials and Methods

### 2.1. Materials

Wheat was purchased from the local commercial market, and the variety was Jimai 22. The water used in this study was manufactured by Milli-Q Ultrapure Water Systems (Shanghai, China). Pepsin from porcine gastric mucosa (>2500 U/mg), trypsin from porcine pancreas (1655 U/mg), and chymotrypsin (>40 U/mg) were purchased from Sigma-Aldrich Chemical Corporation. Unless otherwise specified, all used chemicals were higher than analytical grade. Unless otherwise stated, all chemicals were obtained from Sigma-Aldrich (Shanghai, China).

### 2.2. Protein Sample Preparation

The glutenin used in the experiment was prepared according to the method in our previous paper [12]. Briefly, n-hexane (1:20, n-hexane: gluten; *w*/*v*) was added to gluten and then stirred for 1 h to remove fat, followed by placing the gluten suspension in a fuming cupboard overnight to remove the n-hexane. A 0.4 mol/L NaCl solution (1:20, NaCl: gluten; *w*/*v*) was added to the gluten, followed by stirring again for 1 h. The suspension was then centrifuged to collect the precipitate. Ultrapure water (1:20, *w*/*v*) was added to the precipitate and then stirred for 1 h to remove NaCl, followed by centrifugation to collect precipitate. The precipitate was again dissolved in 70% alcohol (1:40, *w*/*v*) and centrifuged at 10,000× *g* for 20 min to remove the prolamin. Each extraction step was repeated three times.

### 2.3. MGO-Glutenin Preparation

A given mass of glutenin powder was placed in a mortar and fully ground. After grinding to an ultrafine powder, glutenin powder was added to ultrapure water (the mass ratio of glutenin to water was 1:1), which was then homogenized at 10,000 rpm using a high-speed blender (Ika T18 Basic, Staufen, Germany) for several rounds until the glutenin powder was stably suspended in the ultrapure water. In order to make the MGO and glutenin fully react, the mass ratio of MGO and glutenin was selected to be 1:8 (slightly higher than that of MGO and glutenin in actual food). MGO was added to the suspension to provide a mass ratio of glutenin to MGO of 1:8. The suspension was then heated to 100 °C for 15 min to simulate heat processing. The glutenin was mixed with water and underwent the same protocol as MGO-glutenin and served as a control (heated glutenin). Samples were then freeze-dried after ultrafiltration to obtain MGOglutenin.

### 2.4. Structural Characterization of Glutenin

The secondary structure of glutenin and its related reaction products was determined by Fourier transform infrared spectroscopy (FT-IR). A scanning band of 4000–400 cm^−1^ was then used for FT-IR spectroscopy using 32 scanning frames. The corresponding resolution of the spectra was 4 cm^−1^ [13]. The disulfide bond (SS) contents of glutenin were determined using a previously described method [14]. The surface hydrophobicity index (H_0_) of glutenin and its corresponding reaction product was determined by the method published in a former study [15]. The extent of proteolytic hydrolysis (DH%) was determined using methods previously described by Wenjun Wen et al. [16]. All measurements were performed in triplicate.

### 2.5. Determination of Digestibility

In this study, digestibility was determined according to the method in our previous paper [12]. Simulated gastric and intestinal fluids were prepared based on US Pharmacopoeia formulae. The enzyme used in the gastric digestion phase was pepsin (182 U/mg proteins), and the small intestinal digestion phase included trypsin (40 U/mg proteins) and chymotrypsin (0.5 U/mg proteins). The extent of proteolytic hydrolysis (*DH*) was calculated using the following equation:(1)$$DH(\%)=\frac{h_{s}}{h_{total}}\times 100\%$$
where *h_s_* is the concentration (mmol) of free amine groups per gram of protein in the sample and *h_total_* is the concentration (mmol) of free amino groups per gram of protein, assuming complete hydrolysis of the protein (8.83 mmol/g protein). All measurements were made in triplicate.

### 2.6. Mice

Female BALB/c mice aged 6 to 8 weeks were purchased from SiBeiFu Experimental Animal Breeding Co. Ltd. (Beijing, China). All mice used in this study were treated according to the guidelines for the care and use of Laboratory Animals published by the US National Institutes of Health, and all experimental procedures were approved by the Animal Care Review Committee of Tianjin University of Science and Technology. Animals were housed in an air-conditioned room (23 ± 2 °C) with a 12 h light/12 h dark cycle. All mice were allowed free access to food and purified water. All animal experiments began one week after feeding.

### 2.7. Experimental Design

The mice were divided into four groups: three groups of mice sensitized to either native glutenin, heated glutenin, or MGO-glutenin in addition to unsensitized (control group). Mice (*n* = 8) were intraperitoneally sensitized with 10 μg of glutenin mixed on aluminum hydroxide (Sigma-Aldrich, Saint Quentin Fallavier, France) on days 0, 7, 14, 21, and 28. Then, mice were intragastrically administered with 20 mg of glutenin on day 35 [17,18]. The schematic diagram of the experimental design for glutenin-sensitizing of mice is shown in Appendix A.

### 2.8. Allergy Evaluation

First, anaphylaxis symptoms were scored by visually monitoring mice for 1 h after challenge. Anaphylactic symptoms were rated as 0 = no symptoms; 1 = hair up, scratching head and ear; 2 = reduced activity; 3 = swelling around the eyes and mouth; 4 = loss of consciousness, no activity upon prodding; and 5 = convulsion, death.

Blood was then taken from the retro-orbital plexus on day 36 (24 h after intragastric stimulation) and then centrifuged at 3000× *g* for 20 min at 4 °C. The serum was then stored at −80 °C until use. Levels of serum total IgE, histamine, mast cell tryptase (MCT), and serum mouse mast cell protease 1 (mMCP-1) were measured using a commercial ELISA kit according to the manufacturer’s recommendations (NanJingJianCheng Co. Ltd., Nanjing, China).

### 2.9. Cell Separation

GALT was prepared according to the method of Resendiz-Albor A. et al. [19]. The mice were sacrificed by dislocation of cervical vertebra. All small intestines were taken and soaked in pre-cooled D-Hank’s solution, the mesentery was carefully removed, and Peyer’s patches were then carefully cut out and collected. The intestine was repeatedly washed with D-Hank’s solution containing 5% fetal calf serum, and the washed small intestine was cut along the longitudinal axis, placed in a centrifuge tube containing EDTA–DTT (Ethylene Diamine Tetraacetic Acid-dithiothreitol) digest, and then oscillated for 40 min at 180 r/min at 37 °C. The cell suspension was passed through a mesh filter and then collected by centrifugation. The cells were resuspended in 5 mL of 40% Percoll solution, then carefully added to 4 mL of 70% Percoll solution and centrifuged at 1000 r/min for 30 min. The cells collected at the interface were intestinal intraepithelial lymphocytes (IELs). The mesentery and the collected Peyer’s patches (PPs) were ground, filtered with mesh filter, and centrifuged. The cells were resuspended in 5 mL of 40% Percoll solution, carefully added to 4 mL of 70% Percoll solution, and centrifuged at 1000 r/min for 30 min, and the cells at the interface were collected to obtain mesenteric lymph nodes (MLNs) and PPs. Spleens was collected upon sacrifice under sterile conditions. Single-cell suspensions were prepared from spleen by pressing through a cell strainer using a piston, and the collected cells were washed with PBS. To isolate splenocytes, red blood cells were removed by treatment with RBC lysis buffer (Beyotime, Jiangsu, China). The lymphocytes and splenocytes were then used as material for further cytokine assays.

### 2.10. Detection of Cytokines by ELISA

Suspensions of GALT cells (5 × 10^6^ cells/well) were prepared in RPMI-1640 medium containing 100 µg mL^−1^ of either glutenin, heated glutenin, or MGO-glutenin and incubated at 37 °C. The RPMI-1640 medium contained 1% penicillin–streptomycin, 10% fetal bovine serum, 25 mM Hepes buffer, and 5 × 10^5^ M 2-mercaptoethanol. After 24 h, the supernatants of the cultures from each mouse were collected and pooled. Cytokines (IFN-γ, IL-4, IL-10, and TGF-β) in the culture supernatants were determined by commercial ELISA kits (NanJingJianCheng Co. Ltd., Nanjing, China) following the manufacturer’s recommendations.

### 2.11. High-Throughput Sequencing and Bioinformatic Analysis

Isolated fecal DNA was treated as previously reported. Sequencing of 16S rRNA gene amplicons was performed on the Illumina MiSeq platform (Nuohe Zhiyuan Bio-informatics Technology Co. Ltd., Tianjin, China), according to a previous report, to determine sequences of primers targeting the V4 hypervariable region of the bacterial 16S rRNA genes. Bioinformatic analysis of sequencing data was conducted using the Quantitative Insights Into Microbial Ecology (QIIME) software. Briefly, raw data from all samples were filtered and spliced to obtain high-quality clean reads. Taxonomic ranks were assigned to OTU (operational taxonomic unit) representative sequences using Ribosomal Database Project (RDP) Classifier v 2.2. Finally, an OTU table and a phylogenetic tree were produced according to diversity (within sample) and β diversity (between samples) analysis.

### 2.12. Statistical Analysis

Data are presented as mean ± standard deviation (SD) or standard error of mean (SEM) and were analyzed using the SPSS 19.0 software (International Business Machines Corporation, New York, NY, USA). Data were tested by Student’s *t*-test. Differences were considered significant when *p* < 0.05.

## 3. Results

### 3.1. MGO Induced Conformational Changes of Glutenin

To investigate whether MGO modification resulted in changes to the secondary and spatial structure of glutenin, native glutenin, and its related reaction products were analyzed using FT-IR, and the secondary structure was determined using OMICN software based on Fourier transform infrared spectroscopy results. The results show that the unwinding of the α-helix is accompanied by a parallel interaction with the β-sheet, indicating direct transformation of glutenin into a regular structure induced by MGO decoration (Figure 1A). The heat treatment of glutenin without MGO resulted in decreased α-helix structure. In addition, the significant difference in the structural changes of heated glutenin and MGO-glutenin compared with native glutenin was demonstrated by SS and H_0_ analysis, with the heated glutenin showing significantly higher SS content and lower H_0_ (Figure 1C,E). The digestibility of MGO-glutenin was characterized by determining its degree of hydrolysis using the OPA method, and the results show lower DH% of MGO-glutenin compared with heated glutenin (Figure 1D).

### 3.2. MGO-Glutenin Induced a Lower Immune Response than Native Glutenin

Our study evaluated the allergic responses induced by native glutenin, heated glutenin, and MGO-glutenin in a mouse model (Figure 2 and Figure 3). Hypersensitivity symptoms were indicated by scoring from 0 to 5 within 1–1.5 h after each challenge of the mice. All mice in the control group exhibited no hypersensitivity symptoms. Heated glutenin elicited significant hypersensitivity with symptoms of systemic anaphylaxis while mice sensitization with MGO-glutenin showed a lower hypersensitivity reaction (all scores ≤ 1), and three out of 10 were negatively symptomatic. The anaphylactic response score for MGO-glutenin group were relatively discrete. This is because the individual difference is great under this experimental condition, and some mice are more sensitive to MGO-glutenin. Although MGO decoration of glutenin alleviated the resulted allergic reaction, some mice showed reduced activity after the MGO-glutenin challenge, leading to slightly higher allergy scores. (Figure 2). The levels of total IgE and histamine in the serum of MGO-glutenin-stimulated mice were significantly lower than those of the glutenin group (Figure 3A,B). Since the high levels of total IgE and histamine represent a typical allergic response, these results indicate that the sensitivity to MGO-glutenin, in comparison with glutenin, was significantly decreased. The MCT content in mice serum in response to MGO-glutenin was significantly lower than that of the glutenin group, but higher compared with the control group (Figure 3C). No significant difference in the levels of mMCP-1 was observed between the glutenin group and MGO-glutenin group (Figure 3D). The ability of glutenin to induce mast cell degranulation by binding to a specific IgE was significantly reduced by MGO modification but not completely abolished.

### 3.3. MGO-Glutenin Regulated Mouse Immune Response by Inducing Treg Differentiation

To further depict the mechanisms of allergy, the level of cytokines secreted by immune cells was determined based on the supernatants of spleen, thymus, and Peyer’s patch cells. After specific activation by glutenin, the secretion of INF-γ into the cell supernatants of spleen and GALT from glutenin-treated mice was reduced compared with control mice. However, after specific activation of MGO-glutenin, the secretion of IFN-γ in the cell supernatants of GALT from MGO-glutenin-treated mice increased compared with that of glutenin-treated mice (Figure 4A). No such difference was observed in the cell supernatant of spleen and GALT except for MLN cells in heated-glutenin-treated mice. Th2 polarization was indicated by IL-4 secretion, which was found to be increased in the in spleen and GALT of glutenin-treated mice compared to control. IL-4 secretion in spleen and GALT was decreased in heated-glutenin-treated and MGO-glutenin-treated compared with glutenin-treated mice (Figure 4B). As an indicator of Treg activation, the levels of IL-10 and TGF-β in the cell supernatant of spleen and gut-associated tissues were determined. Low levels of TGF-β and IL-10 were found in the cell supernatants of spleen and GALT from the glutenin group, whereas high levels of both cytokines were quantified in the control group. In addition, secretion of TGF-β but not IL-10 was increased in the spleen and GALT from heated-glutenin-treated mice compared to glutenin mice. Stimulated secretion of IL-10 was also observed in GALT from heated-glutenin-treated mice, but no significant difference was observed in spleen (Figure 4C,D).

### 3.4. MGO-Glutenin but Not Glutenin Contributes to the Maintenance of the Beneficial Gut Microflora

The composition of gut microflora was determined by analyzing the abundance of bacteria in feces using 16S genome sequencing to explore the influence of MGO-glutenin on gut microflora. Alpha diversity was used to describe the variation of microbiologic species diversity in the experimental groups. As shown in Figure 5A, the results of ecological indicators showed that the gut microflora richness of glutenin, heated-glutenin, and MGO-glutenin treated mice all increased compared with the control group. Principal coordinate analysis (PCoA) was used to examine the changes in fecal microbiota across groups (Figure 5B). Compared with the glutenin group, a similar PCoA distance between the MGO-glutenin group and control group was observed. To more precisely determine the effect of glutenin and MGO-glutenin on the distribution disparity in gut microflora composition, LEfSe (LDA effect size) analysis was performed to visualize gut microflora abundance at various taxonomic levels. Fifteen taxa with significant differences were verified in four groups (Figure 5C), among which only four taxa with significant differences were observed at the genus level. At the family level, Bacteroidaceae abundance is a significant distinguishing feature of MGO-glutenin challenge, and this is also reflected at the genus level. Another discriminative feature of MGO-glutenin-stimulated mice are *Bacteroides acidifaciens* profiles. In the glutenin group, distinguishing bacteria at the phylum, class, family, and order levels belong to Firmicutes, Clostridia, Ruminococcaceae, and Clostridiales, respectively. In the glutenin group, significant discriminative genera include bacterium_str_77003, proteobacterium UMB8H, and *Actinobacillus*. Thus, *proteobacterium* UMB8H and *Actinobacillus* may be closely related to the development of glutenin-related food allergy.

## 4. Discussion

Protein glycation based on Maillard reaction may lead to structural changes of allergens, thus changing the allergenicity. However, the role of glycation of allergens in regulating immunological properties remains unclear. In this study, the BALB/c mouse model was used to study the effect of MGO decoration of glutenin on its allergic reaction during heat processing. The mechanism of glutenin allergic reaction changes was elucidated by analyzing the structural and digestive changes and the characteristics of gut microflora in mice.

The effect of food processing methods on the allergenicity of allergens is usually dependent on induced structural variations and epitope redistribution as well as alterations to intrinsic biophysical and chemical properties. Moreover, food allergens have the potential to interact with other food matrixes and subsequently play an important role in allergenic regulation [6]. Analysis of the correlation between glycosylation and allergenicity variation indicated an uncertain effect of either increased or decreased allergenicity induced by Maillard reaction during food processing [20]. Nevertheless, reducing the allergenicity of food allergens using glycosylation or the Maillard reaction is a promising strategy that has been studied in related research to determine the correlation between the secondary structure of proteins and the resulting allergic reactions [21]. Furthermore, the loosing of conformational structure during the process of Maillard reaction would increase the possibility to form polymerization and aggregation [22]. Han et al. reported on the hypoallergenic products of the Maillard reaction in which the underlying mechanism of altered allergenicity was induced structural changes [23]. The commonly employed heat-processing strategy of wheat could affect epitope accessibility through induction of aggregation and may irreversibly destroy conformational epitopes. MGO mainly modifies the side chains of Lysine, arginine, histidine, cysteine, tryptophan, and methionine in proteins during heat processing. The allergen epitopes containing these amino acid residues in glutenin might be destroyed and their sensitivity reduced. The identification of the effect of MGO on the epitope of glutenin allergens is of great significance to our experiment. Unfortunately, due to the complex structure of glutenin, it is difficult to identify the change of its epitope in our experiment. FT-IR was used to analyze the secondary structure, and it was found that unwinding of the α-helix in MGO-glutenin may contribute to the decreased allergenicity compared with glutenin. In addition, the increased level of disulfide bonds and reduced surface hydrophobicity could be observed in MGO-glutenin compared with glutenin and heated glutenin. The reduction of hydrophobicity was likely due to the induction of intermolecular crosslinks by MGO that led to the burying of hydrophobic groups within glutenin. It is well known that heating above 55 °C can promote the glutenin disulfide bond/sulfhydryl exchange reaction, thereby promoting the formation of new disulfide bonds [1]. Therefore, the supramolecular structure formed by polymerization and aggregation may appear during the heating process of glutenin [24]. As shown in the Appendix A, the solubility of heated glutenin decreased slightly and the heated MGO-glutenin increased significantly. Martin et al. have shown that disulfide bond production during heat treatment drives the insolubilize of glutenin [25]. High molecular weight glutenin subunits are related to the solubility of glutenin [26]. Early studies proved that fatty acids may bind to cysteine or lysine residues in high molecular weight glutenin subunit, thereby destroying the glutenin structure and changing the solubility of subunits [27,28,29]. In this study, MGO may also destroy the lysine and arginine residues of glutenin, thereby reducing its solubility. However, other factors such as oxidation reaction are not excluded. In general, it is meaningful to further study the mechanism of the solubilization of glutenin by MGO. As a result, a general conclusion could be proposed that MGO alters the structure of glutenin and leads to aggregation. Toheder Rahaman et al. showed that heating to 100 °C induced gliadin aggregation that resulted in decreased digestibility and less availability of antigenic components and therefore minimum antigenicity. This study is similar to our results. Therefore, we speculate that the polymerization and aggregation of glutenin results in the decrease of its digestibility and the decrease of antigen utilization, thus leading to the decrease of antigenicity [30]. In addition, the formation of aggregates may destroy or mask allergen epitopes, thereby preventing IgE binding and cross-linking and subsequent mediator release, which ultimately leads to reduced allergenicity.

It is generally believed that there is a close relationship between protein digestibility and sensitization. In order to trigger an allergic immune response, food proteins (or peptides) must be retained in the gastrointestinal tract to allow sufficient time to induce sensitization. From this point of view, the sensitivity of food proteins to intestinal digestion seems to be an important factor for determining their allergenicity [31]. As shown in Figure 1C, the digestibility of MGO-glutenin was significantly lower than that of glutenin. This is because glycosylated lysine and arginine residues are less susceptible to pepsin/trypsin proteolysis by masking the sites of cleavage. Research shows MGO decoration of protein leads to the destruction of amino acid residues with affinity side chains (such as lysine, arginine, cysteine, and histidine), resulting in reducing its digestibility [32]. It is generally accepted that the resistance of protein to gastrointestinal digestion is an indicator of potential allergic reactions. For instance, comparison of digested and non-digested β-lactoglobulin (β-Lg) in rat models found that non-digested β-Lg induced more IgE and more severe allergic reactions, directly linking β-Lg digestion to allergenicity [33]. In general, high levels of glycosylation will reduce the digestibility of the protein, resulting in increased IgE reactivity of the hydrolysate. However, it has also been demonstrated that aggregate formation leads to increased resistance to protein digestion and reduced sensitization [34]. As in current research, MGO modification resulted in reduced glutenin digestibility and sensitization. This is most probably because the poor absorption of digestive products and the delay in their sensitivity to the immune system lead to a weakened immune response in the mouse model [34].

Next, we studied the effect of MGO modified glutenin on allergic reactions. It will be more meaningful if the serum pool of human allergic to glutenin can be used in the experiment to study the sensitization of glutenin by MGO modification. Taking into account the ethical issues and the convenience of the experiment, we use mice model for this study. The experimental design in this study can also support our conclusions. As concluded, our study shows that MGO decoration of glutenin results in reduced sensitization toward glutenin as demonstrated using a mouse model. First, MGO-glutenin-sensitized mice were observed with relatively minor hypersensitive symptoms and a lower level of total IgE compared with glutenin-sensitized and control mice. The consistent variation in allergic symptom score and IgE levels reflects the difference in the degree of sensitization for MGO-glutenin. The degranulation of mast cells occurs during the effector phase of the food allergy, which can reflect the ability of antigens to stimulate allergic reactions. In the case of the re-interaction with the antigen, mast cells degranulate and release effectors such as histamine directly related to the pathological clinical symptoms of food allergy [35]. In this study, it was also observed that the level of histamine in glutenin-sensitized mice was elevated compared with control mice, was restored in MGO-glutenin allergic mice. This result further suggests the lower immune response induced by MGO-glutenin. The development of allergic reactions is accompanied by the release of trypsin by mast cells. The increased activity of mast cell trypsin in mice is due to the release of mast cell trypsin in secretory granules following the activation of mast cells. Mast cell trypsin can destroy the integrity of the relevant tissue membrane, promoting tissue remodeling and the progression of inflammation as induced allergic symptoms. Therefore, MCT is a more selective marker of mast cell-mediated inflammatory response [36]. However, a low level of MCT was observed in MGO-glutenin-sensitized mice, which suggested the reduced sensitization potential resulting from MGO decoration in BALB/c mice. No significant difference in mMCP-1 contents was observed between the glutenin group and MGO-glutenin group. As one of the bioactive substances released after mast cell degranulation, mMCP-1 can be used as the indicator of this process [37]. Though no significant difference in mMCP-1 levels was verified between the glutenin group and MGO-glutenin group to reflect the difference of the effect stage after activation, mMCP-1 levels in the MGO-glutenin group still showed a downward trend compared with the glutenin group. In general, glycosylation of food allergens may alter their immunological behavior.

Correlation analysis was performed to unveil the relationship between structure change and sensitization of glutenin. As shown in Figure 1B, α-helix structure and H_0_ are positively correlated with IgE and MCT, while β-sheet formation is negatively correlated with mMCP-1. The change in in protein secondary structure caused by glycation is an effective strategy to mask epitopes related to the allergenicity of food allergens. Several studies have demonstrated the suppressed allergic response resulting from the alteration of secondary structure [38,39]. Gupta R.K. et al. demonstrated that reduced α-helix structure induced by glycation may shield the epitopes of protein and lead to a reduction in allergenicity compared to native chickpea albumin [8], which is consistent with the results of this study. Otherwise, changes in disulfide bonds are also an important factor related to the epitope variation. Mameri, H. et al. demonstrated that the binding ability of gliadin to IgE was reduced due to disulfide bond changes under heating conditions [40]. Similarly, in this study, the presence of disulfide bonds was negatively correlated with MCT, mMCP-1, and His levels. We have recently demonstrated that glutenin can form aggregates through hydrophobic interactions as a result of MGO modification [12]. The formation of aggregates may prevent binding between antigen and antibody epitopes. Moreover, cross-linking of proteins seems to reduce epithelial uptake, which has been demonstrated using crossed-linked β-Lg [39]. In addition, the larger agglomerates can be further metabolized by intestinal microbes, resulting in the formation of new bioactive compounds with the function of modulating the gut microflora composition [41]. Epitope modification induced by the Maillard reaction may affect the allergenicity of wheat protein. Lysine, arginine, histidine, cysteine, tryptophan, and methionine may be involved in the Maillard reaction between MGO and glutenin [42], which facilitates the conclusion that glutenin epitopes containing the above amino acids might be destroyed to reduce their affinity for IgE binding, resulting in reduced sensitization toward glutenin. The above results indicate that MGO can alter the secondary and tertiary structures of glutenin under heating conditions, thereby affecting epitope accessibility by inducing aggregation and resulting in irreversible conformational epitope destruction. It seems to be of special relevance to consider that impaired enzymatic protein digestion is associated with enhanced allergenicity of food proteins. The reason is that reduced digestive capacity results in larger protein fragments that are recognized by the cells of the immune system [43]. However, in this study, DH% is positively correlated with IgE. This may be because of the poor absorption of digestive products and a delay in their sensitivity to the immune system. In addition, longer peptides could alter the composition of the microbiome [44]. In other words, the gut microflora could degrade MGO-glutenin and produce hypoallergenic glutenin peptides.

The interaction between GALT cells and non-immune cells is important for the maintenance of immune tolerance. A previous study has shown that oral tolerance to ovalbumin cannot be induced in PP-deficient mice, which suggests the essential role of PPs in the mucosal immune response [45]. There is clear evidence that MLNs play a key role in inducing mucosal immunity or tolerance. Within hours of the protein entering the intestine, antigen recognition occurred in MLNs, and natural T-cell activation and division took place primarily in MLNs. Previous studies have shown that Tregs can reduce the occurrence of allergies against food allergens, and Treg induction may be the basis for the protective effect of certain dietary interventions in the food allergy model [46]. Transforming growth factor-β (TGF-β) and IL-10 are Treg-related cytokines that are essential for maintaining immune tolerance and reducing allergic reactions [47]. MGO-glutenin leads to increased TGF-β and IL-10, which suggests that MGO-glutenin may induce Treg differentiation. In addition, there is increasing evidence that oral tolerance is mediated through immunosuppressive activation in the gut. The main cells involved in this process are Tregs, which are derived from T-cells after exposure to allergens in the presence of TGF-β [48]. IFN-γ and IL-4 are cytokines released by Th1 and Th2 cells, respectively. Under normal conditions, Th1/Th2 cells are in a relatively balanced state in mice. When dysfunctions such as allergic reactions occur, the balance shifts to Th2 cells, leading to a range of symptoms, including ear swelling, IgE increase, and mast cell threshing [45]. The increased IL-4 levels and the decreased IFN-γ levels in the supernatant of all cells in the MGO-glutenin group indicate that MGO-glutenin inhibited allergic reactions by inhibiting Th2 cell differentiation. These effects coincide with the lower production of Th2-related cytokines, which might dampen Th2 response. This means that MGO-glutenin inhibits Th2 cell differentiation and participates in the induction of beneficial effects by inducing regulatory T-cells and has a profound effect on tolerance maintenance.

The changes of gut microflora in mice were studied by 16S sequencing, which could only reflect the relative changes of flora abundance. To study the effect of MGO-glutenin on the composition of the gut microflora, we used PCoA to compare the glutenin group, heated-glutenin group, and MGO-glutenin group. The PCoA showed marked difference in gut microflora between the glutenin group and MGO-glutenin group. Thus, it was suggested that MGO-glutenin had a significant effect on the intestinal microbes in mice. LEfSe analysis was used to detect the key taxa that differ between the four groups, so as to compare the relative contributions of the discrepant taxa. A total of 15 taxa of different levels were identified to have significant abundance differences across the four groups. In our study, the most significant difference between the MGO-glutenin and other groups in terms of gut microflora composition was associated with the phylum Bacteroidetes, especially for the family Bacteroidaceae and the genus *Bacteroides*, whose presence is an important characteristic of the MGO-glutenin group. *Bacteroides* has been identified to be conducive to promoting the development of Treg, and to promote tolerance to dietary antigens by inducing the expression of transcription factor RORgt in nascent Treg cells through the upstream myd88-dependent mechanism. Our results provide some interesting insights into the relationship between changes of gut microflora caused by protein glycation and immune response regulation. In future, we plan to conduct studies to verify that *Bacteroides*, which is characteristic of the MGO-glutenin group, participates in immune regulation. Bacteroidetes are thought to be involved in metabolic transformation, usually associated with protein degradation, which is essential for the host. Unlike other allergen proteins, gluten is rich in proline and glutamine residues, which are exceptionally resistant to enzyme degradation in mammalian digestive tracts [49]. This incomplete digestion facilitates the production of longer oligopeptides for the interaction with antigen-presenting cells, which can activate the T-cell response associated with wheat protein allergy [50]. Bacteroidetes can metabolize the digested glutenin into small molecules without associated immunogenicity. Recent studies have shown that symbiotic microorganisms that colonize the intestinal tract have a strong regulatory effect on Th2 immune responses [51]. An obvious example was the observation that in the absence of microorganisms, mice that had not been treated with antibiotics were susceptible to allergy and had elevated levels of IL-4, basophils, and serum IgE, thereby enhancing the Th2 immune response, which suggested that microorganisms are important players for the modulation of Th2 immune response [52]. The potential pathway may depend on the mediation of gut microbes to regulate the differentiation of induced Tregs, thereby suppressing the Th2 immune response. Ohnmacht’s research showed that the gut microbes stimulated the expression of RORγt in Tregs and inhibited the Th2 cells, so as to avoid the formation of IL-4 and IgE. This suggested that in the MGO-glutenin group, Bacteroidetes, promote the differentiation of Tregs, which may function in suppressing the Th2 response. Considering the effects of Tregs and Th2 cells on allergic diseases, mucosal immunity, and intestinal flora regulation, a correlation could be established between the immune response and the gut microflora through the determined increased level of Tregs and decreased Th2 cells in the MGO-glutenin group, with further evidence that Bacteroidetes alleviated the immune response in the MGO-glutenin group. Similar to our results, Caminero A. et al. reported a reduction in specific bacterial populations such as *Lactobacilli* and *Bacteroides* that metabolize gluten in celiac disease patients [53]. The most significant differences between the glutenin group and other groups at phylum, family, and genus level were the phylum Bacteroidetes, the family Ruminococcaceae, and the genus *Actinobacillus*. Ruminococcaceae is positively correlated with inflammation-related diseases [54]. In this study, *Actinobacillus* was the only genus of the biological marker in the glutenin group, which has not been previously reported. Therefore, the relative abundance of *Actinobacillus* may be closely related to the development of glutenin-related food allergy. Pearson correlation analysis showed that the abundance of actinomycetes was positively correlated with Th2 cytokines and negatively correlated with Th1 cytokines, which indicates that actinomycetes may be involved in the promoted Th2 cell differentiation, but the detailed mechanism remains to be elucidated. The current research demonstrates the reduced immune response of MGO-glutenin compared with glutenin, which mainly depends on the protective effect of *Bacteroides* in promoting Treg differentiation and inhibiting Th2 differentiation.

In summary, our study proved the previous hypothesis that MGO decoration of glutenin would alleviate allergic reactions in mice. MGO decoration may contribute to the aggregation of glutenin caused by conformational changes in the secondary and tertiary structures, which has the potential to mask or even destroy surface epitopes and mitigate sensitization. In addition, MGO-glutenin alters the composition of gut microflora. *Bacteroides*, which may be a marker microorganism in the feces of MGO-glutenin-sensitized mice, functions in inducing the polarization of Tregs to facilitate the stimulation of immune tolerance and inhibition of the Th2 immune response as part of a general effect that dampens the immune response.

## 5. Conclusions

This study is the first to investigate the effect of MGO decoration of glutenin on the resulting allergic reaction in mice during heat processing. The current research results show that MGO decoration of glutenin could alleviate the resulting allergic reaction in mice. This remission is achieved by changing the structure and digestibility of glutenin and the intestinal flora of mice. This study provides a theoretical basis for alleviating glutenin allergic reactions through processing. However, the mechanism by which *Bacteroides* challenged mice with MGO-gluten induces Treg cell polarization and suppresses Th2 immune response needs to be further elucidated.

## Figures and Tables

**Figure 1 nutrients-12-02844-f001:**
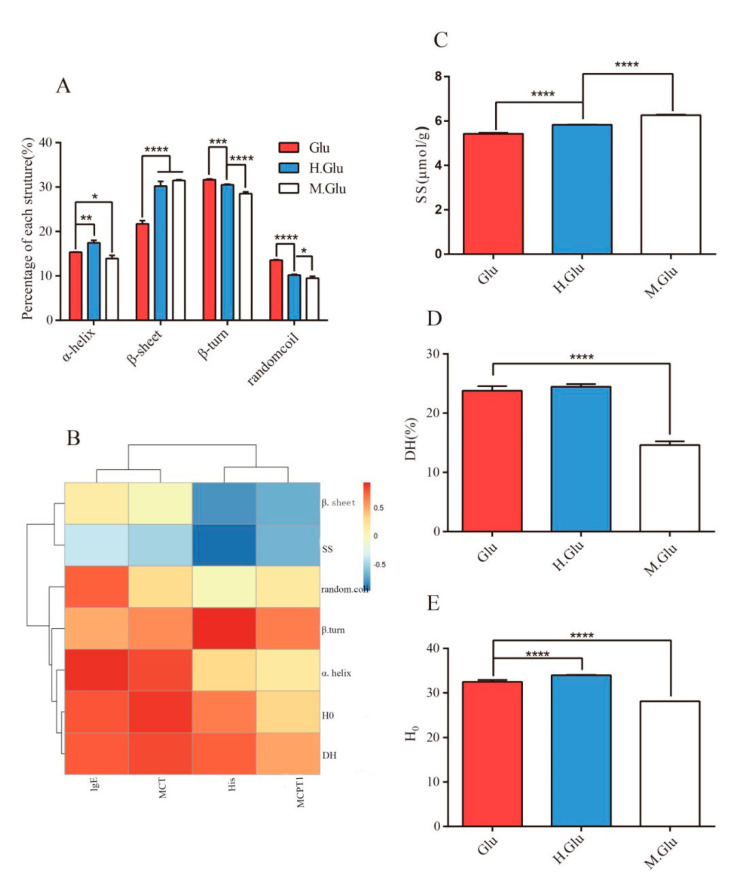
Methylglyoxal (MGO) changes the structure of glutenin. (**A**) Secondary structure changes. Significant differences among α-helix, β-sheet, β-turn, and random coil were indicated as * *p* < 0.05, ** *p* < 0.01, *** *p* < 0.001, **** *p* < 0.0001. (**B**) Correlation analysis between structural changes and sensitivity of glutenin. (**C**) Disulfide bond (SS) levels of glutenin and its corresponding reaction products. (**D**) Proteolytic hydrolysis (DH%) levels of glutenin and its corresponding reaction products. (**E**) H_0_ levels of glutenin and its corresponding reaction products. Glu, H.Glu, and M.Glu represent native glutenin, heated glutenin, and MGO-glutenin, respectively.

**Figure 2 nutrients-12-02844-f002:**
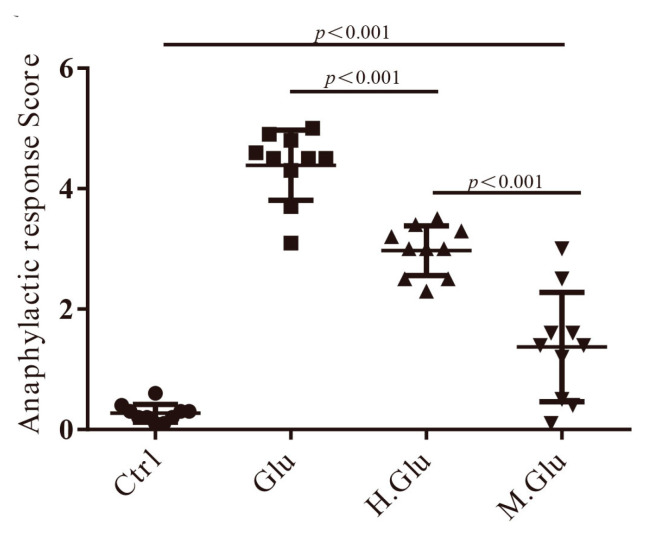
Hypersensitivity symptom scores of mice. Mice sensitization with MGO-glutenin showed a low hypersensitivity reaction (all scores ≤ 1). BALB/c mice were treated with normal saline (control group), glutenin, heated glutenin, and heated MGO-glutenin. Ctrl represents the control group and Glu, H.Glu, and M.Glu represent native glutenin, heated glutenin, and MGO-glutenin, respectively. The triangle represents the control group, the square represents native glutenin, the upward triangle represents heated glutenin, and the downward triangle represents MGO-glutenin.

**Figure 3 nutrients-12-02844-f003:**
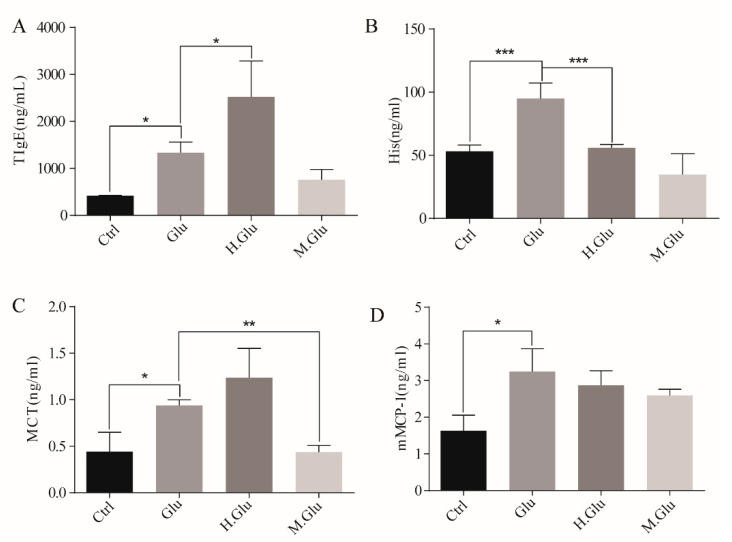
MGO-glutenin induced a lower immune response than glutenin. Ctrl represents the control group and Glu, H.Glu, and M.Glu represent native glutenin, heated glutenin, and MGO-glutenin, respectively. The bars indicate the serum levels of total IgE (**A**), His (**B**), MCT (**C**), and mMCP-1 (**D**). Data are presented as mean ± SEM. Significant differences of Glu versus H.Glu and M.Glu groups are indicated as * *p* < 0.05, ** *p* < 0.01, *** *p* < 0.001.

**Figure 4 nutrients-12-02844-f004:**
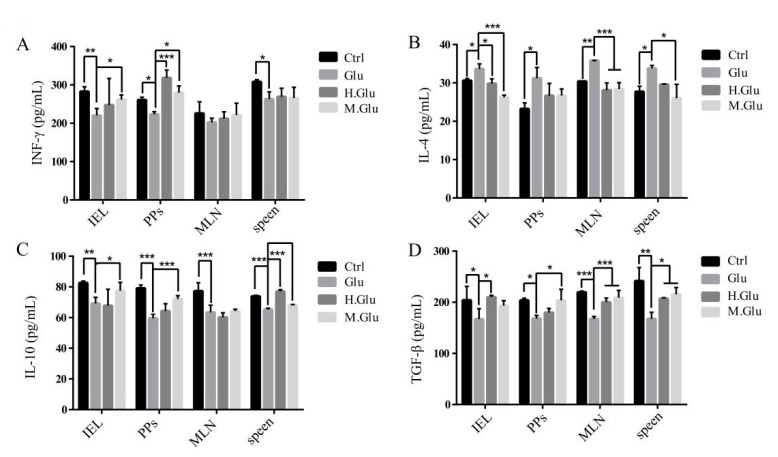
MGO-glutenin regulated mouse immune response by inducing Treg differentiation. The production of IFN-γ (**A**), IL-4 (**B**), IL-10 (**C**), and TGF-β (**D**) cytokines measured by ELISA in the supernatant from lymphoid cells. Ctrl represents the control group and Glu, H.Glu, and M.Glu represent native glutenin, heated glutenin, and MGO-glutenin, respectively. Significant differences of Glu versus H.Glu and M.Glu group are indicated as * *p* < 0.05, ** *p* < 0.01, *** *p* < 0.001. IEL: intestinal intraepithelial lymphocytes, PPs:, MLN: mesenteric lymph nodes.

**Figure 5 nutrients-12-02844-f005:**
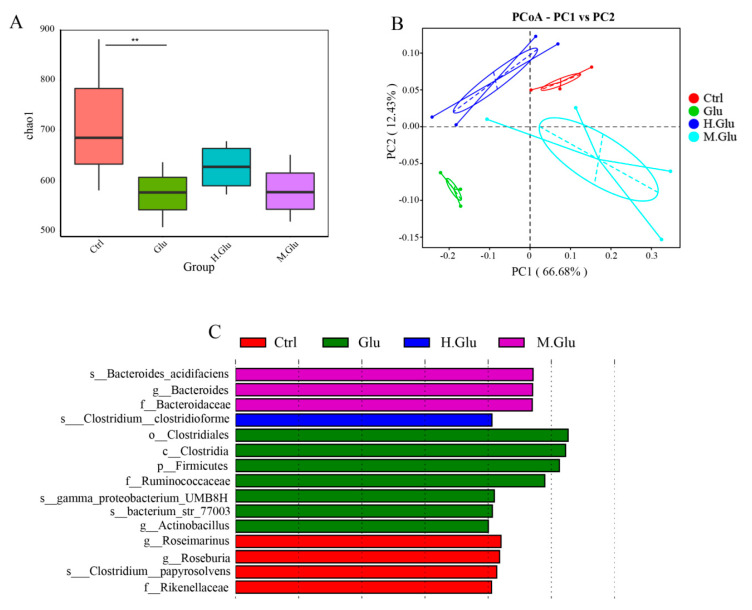
MGO regulates immune response by regulating the microbial composition of mice feces. Ctrl represents the control group and Glu, H.Glu, and M.Glu represent native glutenin, heated glutenin, and MGO-glutenin, respectively. Genomic DNA was extracted from the fecal samples taken from mice just before sacrifice. (**A**) Microbiota diversity of each group. (**B**) Principal coordinate analysis (PCoA) of each group. (**C**) LEfSe analysis microbiota diversity of each group. ** *p* < 0.01.

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
