# Peer review of "Methylglyoxal Decoration of Glutenin during Heat Processing Could Alleviate the Resulting Allergic Reaction in Mice"

_nutrients, 2020, doi:10.3390/nu12092844_

Round 1
Reviewer 1 Report
In the manuscript “Methylglyoxal decoration of glutenin during heat processing could alleviate the resulted allergic reaction in mice” Wang Yaya and co-authors examined the impact of the use of MGO, one of the most reactive molecules produced during the transformation process, on gluten hypersensitivity, especially glutenins. This work is the continuation of a previous publication dealing with the aspects of protein solubilization. The article is structured around 3 axes. First, the authors are interested in structural modifications, in particular the secondary structure of glutenins. In-Vitro hydrolysis has also been discussed. Then, the authors used mice as a model for the study of allergenicity of wheat proteins. Finally, the gut microflora was studied to determine the impact of MGO-modified glutenins.
The article is well written. However, the manuscript needs proofreading. Some references are not numbered in the text. Some sentences could be simplified. Results on solubility could have been exploited in this paper.
Further remarks concern :
Line 94: Chemical are purchased from Sigma alrich USA!!!, Milli-Q from France!!!. hese companies exist in china! Asia!
Line 98: please check if there is a need to put space before the citation number! Please check all references
Line 102: was instead of is!
Line 104: Could the authors explain their choice of the glutenin to MGO 1:8 ratio?
Line 105: Were the controls (heated-glutenin) also mixed with water and underwent the same protocol as MGO-glutenin.
Line 115: please check ref citation number!
Line 139: space before µg!
Line 139: Is “absorbed” the right adj
Line 154: dot after al.
Fig2: any explanation for the dispersion of anaphylactic response score for MGO-glutenin group
Line 270 to 272: I did not find that MGO-glutenin group partially restored the gut microflora richness!!!
Line 285: The sentence must be in the conditional tense. Several studies (on different matrices) should be conducted to confirm your results.
Line 286-288: what do you mean by this sentence!
Fig 5B: groups and legends are not readable!
Line 306: there were polymerization and or aggregation!! What were the effects of Maillard reaction on glutenins? How can authors prove the aggregation phenomenon?
Line 306: pleade add et al. !!!
Line 309: what about positional epitopes?
Line 312: How can authors explain the reduction of hydrophobicity?
Line 313-315: aggregation occurred in seed and wheat grain transformations!!! Polymerization and aggregation are tightly related!
Line 316: we talk about a three-dimensional structure...glutenins have a quaternary structure. Please change “the tertiary”!
Line 317: relationship between native glutenin aggregates and allergenicity has been recently reported!
Line 327: Other amino acids may interact with the MGO. Please check and correct.
Line 368: please check Gupta Rk (et al. is missing), citation number!!?
Line 370: chickpea albumin8: 8 is the name/classification!
Line 375: please check reference! Add [ ]
Line 381: again; other amino acids may be involved!
Line 445: number citation!!
Line 459: Are there any references that can support these statements?
Line 469: it is judicious to also talk about the conformation of glutenins or aggregates.
Author Response
Dear Editors and Reviewers:
Thank you very much for your letter and comments on our manuscript entitled “Methylglyoxal decoration of glutenin during heat processing could alleviate the resulted allergic reaction in mice” (ID: nutrients-924597). We greatly appreciate the constructive comments that are very helpful for our revision of the manuscript. We have made additions and corrections according to your comments. The manuscript was revised by the English editing service from MDPI to make the language of the manuscript more concise and accurate. The revised sections in the manuscript are highlighted in red.
We do hope that the revised manuscript adequately addressed your comments. The major corrections and the responses to the reviewers’ comments are as follows:
General comments: In the manuscript “Methylglyoxal decoration of glutenin during heat processing could alleviate the resulted allergic reaction in mice” Wang Yaya and co-authors examined the impact of the use of MGO, one of the most reactive molecules produced during the transformation process, on gluten hypersensitivity, especially glutenins. This work is the continuation of a previous publication dealing with the aspects of protein solubilization. The article is structured around 3 axes. First, the authors are interested in structural modifications, in particular the secondary structure of glutenins. In-Vitro hydrolysis has also been discussed. Then, the authors used mice as a model for the study of allergenicity of wheat proteins. Finally, the gut microflora was studied to determine the impact of MGO-modified glutenins.
The article is well written. However, the manuscript needs proofreading. Some references are not numbered in the text. Some sentences could be simplified. Results on solubility could have been exploited in this paper.
Response: Thanks for your valuable suggestions. The solubility of glutenin has been supplemented on Fig. S2. in the supporting information. As shown in the figure S2, the solubility of heated glutenin decreased slightly and the heated MGO-glutenin increased significantly. Martin et al. have shown that disulfide bond production during heat treatment drives the insolubilize of glutenin[1]. High molecular weight glutenin subunits are related to the solubility of glutenin[2]. Early studies proved that fatty acids may bind to cysteine or lysine residues in HMW-GS, thereby destroying the glutenin structure and changing the solubility of subunits[3-5]. In this study, MGO may also destroy the lysine and arginine residues of glutenin, thereby reducing its solubility. However, other factors such as oxidation reaction are not excluded. In general, it is meaningful to further study the mechanism of the solubilization of glutenin by MGO. We will continue to explore this issue in subsequent experiments. Information on changes in glutenin solubility has been supplemented in the manuscript. (line 337-345)
Fig. S2. Solubility of native glutenin, heated glutenin, and heated MGO-glutenin samples.
References for responses to general comments
[1] Martin, C.; Morel, M.-H.; Reau, A.; Cuq, B. Kinetics of gluten protein-insolubilisation during pasta processing: Decoupling between time- and temperature-dependent effects. Journal of Cereal ence 2019, 88.
[2] Huang, D.Y.; Khan, K. Quantitative determination of high molecular weight glutenin subunits of hard red spring wheat by SDS-PAGE. I. Quantitative effects of total amounts on breadmaking quality characteristics. Cereal Chemistry 1997, 74.
[3] Rousselphilippe, C.; Pina, M.; Graille, J. Chemical lipophilization of soy protein isolates and wheat gluten. European Journal of Lipid Science and Technology 2000, 102, 97-101.
[4] Pommet, M.; Redl, A.; Morel, M.; Guilbert, S. Study of wheat gluten plasticization with fatty acids. Polymer 2003, 44, 115-122.
[5] Wilkes, M.A.; Copeland, L. Storage of Wheat Grains at Elevated Temperatures Increases Solubilization of Glutenin Subunits. Cereal Chemistry 2008, 85, 335-338.
Point 1: Line 94: Chemical are purchased from Sigma alrich USA!!!, Milli-Q from France!!!. these companies exist in china! Asia!
Response 1: Thank you for your valuable suggestion. The company addresses of Sigma-Aldrich and Milli-Q have been modified to Shanghai, China. (line 88)
Point 2: Line 98: please check if there is a need to put space before the citation number! Please check all references
Response 2: We have put a space before the citation number and carefully checked and corrected all references. (line 91)
Point 3: Line 102: was instead of is!
Response 3: Thank you for your correction. We have replaced “is” with “was” and carefully checked and corrected the tenses of all the sentences. (line 102)
Point 4: Line 104: Could the authors explain their choice of the glutenin to MGO 1:8 ratio?
Response 4: In this experiment, we established a simulated reaction system of MGO and glutenin. The content of MGO in bread is about 60mg/kg, and the content of gluten is about 2g/kg. In this experiment, in order to make MGO fully react with glutenin, we chose a higher ratio than the actual food. We have added instructions to the manuscript to make it easier to understand for the reader.(line 104-105)
Point 5: Line 105: Were the controls (heated-glutenin) also mixed with water and underwent the same protocol as MGO-glutenin.
Response 5: The controls were also mixed with water and underwent the same protocol as MGO-glutenin. We have changed the vague statement to “The glutenin was mixed with water and underwent the same protocol as MGO-glutenin and served as controls (heated-glutenin)”. (line 107-108)
Point 6: Line 115: please check ref citation number!
Response 6: Thanks for your comments. The reference number has been checked. (line 118)
Point 7: Line 139: space before µg!
Response 7: Thank you for your correction. We have added a space before µg. (line 140)
Point 8: Line 139: Is “absorbed” the right adj
Response 8: Thank you for your correction. We have replaced “absorbed” with “mixed”. (line 140)
Point 9: Line 154: dot after al.
Response 9: Thank you for your correction, we have added a dot to “al”. (line 155)
Point 10: Fig2: any explanation for the dispersion of anaphylactic response score for MGO-glutenin group
Response 10: Thank you for your kind reminding. The reason for the dispersion of anaphylactic response score for MGO-glutenin group is that individual difference is great under this experimental condition, and some mice are more sensitive to MGO-glutenin. Although MGO decoration of glutenin alleviated the resulted allergic reaction, some mice showed reduced activity after the MGO-glutenin challenge, leading to slightly higher allergy scores. The explanation for the dispersion of anaphylactic response score for MGO-glutenin group have been supplemented in the manuscript. (line 224-229)
Point 11: Line 270 to 272: I did not find that MGO-glutenin group partially restored the gut microflora richness!!!
Response 11: Thank you very much for your kind reminder. We apologize for the wrong description of the experimental data. We have revised the wrong description to " As shown in Figure 5A, the results of ecological indicators showed that the gut microflora richness of glutenin, heated-glutenin and MGO-glutenin treated mice were all increased compared with the control group" and discussed accordingly. (line 279-281)
Point 12: Line 285: The sentence must be in the conditional tense. Several studies (on different matrices) should be conducted to confirm your results.
Response 12: We gratefully appreciate your valuable suggestions. We have changed the sentence into conditional tense. In follow-up research, Proteobacterium and Actinobacillus will be cultured by selective glutenin medium to explore whether they are indicator microorganisms for glutenin intake. (line 294-295)
Point 13: Line 286-288: what do you mean by this sentence!
Response 13: This sentence had no practical meaning in this section. It was just the prospect of our research work, which was intended to show that we would pay attention to the genera highly relevant with glutenin-induced allergy and MGO-glutenin regulation of allergy. Therefore, we have deleted this sentence, and related views have been added in the conclusion. (line 524-525)
Point 14: Fig 5B: groups and legends are not readable!
Response 14: Thank you for your correction. We have modified the groups and legends and redrawn Figure 5.
Point 15: Line 306: there were polymerization and or aggregation!! What were the effects of Maillard reaction on glutenins? How can authors prove the aggregation phenomenon?
Response 15: Thank you very much for your correction. In this study, polymerization and aggregation co-existed during the reaction between MGO and glutenin. MGO induceed intermolecular cross-linking which might cause the hydrophobic groups in glutenin to be buried and then the hydrophobicity decreased. The SS of glutenin was broken and the subunits were rearranged during the heating process. In addition, MGO had a double carbonyl structure, which might lead to the formation of cross-linked structures between glutenin. These chemical changes were polymerization processes. However, flocculent precipitation could be observed during heating of MGO and glutenin. The formation of precipitation is often related to protein aggregation. During the heating process, the incompletely denatured glutenin might aggregate due to dehydration.
Our study showed that the Maillard reaction between MGO and glutenin resulted in the change of glutenin structure, including the decrease of disulfide bond, the change of secondary structure, the decrease of hydrophobicity, and finally the aggregation of glutenin. In addition, the Maillard reaction results in a decrease in glutenin digestibility.
The manuscript has been revised to avoid misunderstanding. In addition, we have added some descriptions to discuss the changes in the structure of glutenin caused by MGO modification. (line 319)
Point 16: Line 306: pleade add et al. !!!
Response 16: Thank you for your correction, we have added et al. and carefully checked and corrected all references. (line 319)
Point 17: Line 309: what about positional epitopes?
Response 17: This study aimed to determine the effect of MGO on the allergenicity of glutenin based on the Balb/c mouse model pre-sensitized to native glutenin, heated glutenin and MGO-glutenin, as well as elucidate the detail mechanism behind the reduced potential of allergic reaction resulted by MGO decoration. Therefore, a mixture with different types of high-molecular-weight glutenin subunits and low-molecular-weight glutenin subunits, which are representative of the complete structure of glutenin, were selected as the study object. We have tried to study the effect of MGO on glutenin epitopes by HPLC-MS/MS. Unfortunately, due to the complex structure of the glutenin used in our laboratory, we were unable to detect its epitope changes successfully.
According to the literature, MGO mainly modifies the side chains of lysine and arginine in proteins during thermal processing. The glutenin allergen epitope contains lysine residues, so we can speculate that modification in MGO will destroy the glutenin allergen epitope and reduce its allergenicity.
We have added discussion to the manuscript to make it more complete and accessible to readers. In follow-up research, we plan to obtain the subunits of glutenin through separation and purification, and then use mass spectrometry to study the effect of MGO on glutenin epitopes. (line 323-326)
Point 18: Line 312: How can authors explain the reduction of hydrophobicity?
Response 18: The reduction of hydrophobicity was likely due to the induction of intermolecular crosslinks by MGO that led to the burying of hydrophobic groups within glutenin. The lack of hydrophobicity is generally associated with intramolecular or intermolecular aggregation of proteins that affect the affinity of ANS for protein surfaces, or bury hydrophobic sites within protein molecules. In addition, the covalent linkage of MGO to the amino groups of Lys residues increases the net negative charge and hydroxyl groups on the surface of glutenin. These processes may lead to disruption of hydrophilic/hydrophobic balances on protein surfaces. We added an explanation for the reduction of hydrophobicity in the manuscript. (line 332-334)
Point 19: Line 313-315: aggregation occurred in seed and wheat grain transformations!!! Polymerization and aggregation are tightly related!
Response 19: Thanks for your valuable comments. The manuscript has been revised to avoid misunderstanding. (line 334-337)
Point 20: Line 316: we talk about a three-dimensional structure...glutenins have a quaternary structure. Please change “the tertiary”!
Response 20: Thank you very much for your correction. We have changed “the tertiary” to “the structure”. (line 346)
Point 21: Line 317: relationship between native glutenin aggregates and allergenicity has been recently reported !
Response 21: Thanks for your kind comments. We reviewed the literature on the relationship between protein aggregation and allergenicity. We found that the relationship between gliadin and allergenicity has been reported. Unfortunately, no relevant literature about glutenin. Toheder Rahaman et al. showed that heating to 100 °C induced gliadin aggregation that resulted in decreased digestibility and less availability of antigenic components and therefore minimum antigenicity[6]. This study is similar to our results. Therefore, we speculate that the polymerization and aggregation of glutenin results in the decrease of its digestibility and the decrease of antigen utilization, thus leading to the decrease of antigenicity. In order to make our manuscript more complete, we have cited relevant literature on the relationship between gliadin and allergenicity, and revised and improved the relevant expressions. (line 347-351)
Reference for Response 21
[1] Toheder Rahaman; Todor Vasiljevic; Lata Ramchandran. Effect of heat, pH and shear on digestibility and antigenic characteristics of wheat gluten. European Food Research & Technology 2016.
Point 22: Line 327: Other amino acids may interact with the MGO. Please check and correct.
Response 22: Thank you for your correction. MGO can easily modify amino acid residues with nucleophilic side chains (such as lysine, arginine, cysteine and histidine). Sanja Milkovska-Stamenova et al. identified the modified products of arginine, lysine and cysteine in milk by MGO[1]. Research by Jasmin Meltretter et al. showed that there are oxidative modifications of methionine, tryptophan and cysteine in milk during heating, and α-dicarbonyl compounds contribute to the oxidation of amino acids in milk[2]. We have revised the relevant description in the manuscript. (line 360-362)
Reference for Response 22
[1] Milkovska-Stamenova, S.; Mnatsakanyan, R.; Hoffmann, R. Protein carbonylation sites in bovine raw milk and processed milk products. Food Chemistry 2017, 229, 417.
[2] Meltretter, J.; Wüst, J.; Pischetsrieder, M. Modified Peptides as Indicators for Thermal and Nonthermal Reactions in Processed Milk. Journal of Agricultural & Food Chemistry 2014, 62, 10903.
Point 23: Line 368: please check Gupta Rk (et al. is missing), citation number!!?
Response 23: Thank you for your correction. We have added “et al.”after “Gupta Rk”, and supplemented reference and its citation number. (line 407)
Point 24: Line 370: chickpea albumin8: 8 is the name/classification!
Response 24: We have removed “8” and corrected the writing. (line 409).
Point 25: Line 375: please check reference! Add [ ]
Response 25: Thank you for your correction. We have added “[ ]” and carefully checked and corrected all references. (line 415)
Point 26: Line 381: again; other amino acids may be involved!
Response 26: Thank you for your correction. We have revised the manuscript and added other amino acids. (line 420-424)
Point 27: Line 445: number citation!!
Response 27: Thank you for your correction. Supplemented reference and its citation number have been added. (line 486)
Point 28: Line 459: Are there any references that can support these statements?
Response 28: We have added references at corresponding locations. (line 500)
Point 29: Line 469: it is judicious to also talk about the conformation of glutenins or aggregates.
Response 29: Thank you very much for your kind reminder. It is well known that heating above 55°C can promote the glutenin disulfide bond/sulfhydryl exchange reaction, thereby promoting the formation of new disulfide bonds[1]. Therefore, the supramolecular structure formed by intermolecular aggregation may appear during the heating process of glutenin. We have supplemented the relationship between glutenin conformation of glutenin and aggregation caused by heat processing. (line 334-337)
Reference for Response 29
[1] Martin, C.; Morel, M.-H.; Reau, A.; Cuq, B. Kinetics of gluten protein-insolubilisation during pasta processing: Decoupling between time- and temperature-dependent effects. Journal of Cereal ence 2019, 88.

Reviewer 2 Report
This Original Article is ver interesting and demonstrate experimentally in mice the great efficacy of the Methylglioxal decoration of glutenin during the heat procession that could alleviate the resulted allergic reaction in rats.
Trough several very nice experiments performed in different situations and compared with a control group the authors obtain clear differences in the mode used for the prevention and treatment of these reactions in mice
I send some recommendations mainly directed for improving the final redaction of the manuscript

Author Response
Response to Reviewer 2 Comments
Dear Editors and Reviewers:
Thank you very much for your letter and comments on our manuscript entitled “Methylglyoxal decoration of glutenin during heat processing could alleviate the resulted allergic reaction in mice” (ID: nutrients-924597). We greatly appreciate the constructive comments that are very helpful for our revision of the manuscript. We have made additions and corrections according to your comments. The manuscript was revised by the English editing service from MDPI to make the language of the manuscript more concise and accurate. The revised sections in the manuscript are highlighted in red.
We do hope that the revised manuscript adequately addressed your comments. The major corrections and the responses to the reviewers’ comments are as follows:
Point 1: Abstract
This will be written in a “structured form” including the corresponding sections of : Background, Methods, Results and Conclusion.
Response 1: We greatly appreciate your valuable suggestions. We have revised the Abstract.(line 15-30)
Point 2: Keywords
They are good enough and well selected
Response 2: We greatly appreciate your valuable suggestions. We have modified the inappropriate keywords.(line 31)
Point 3: Introduction
Is too long. I want to recommend shorten some redundant information and limit to do a brief description of two main items
1/. The problem
2/. The proposed solution
You probably need two or three sentences for your introduction, but it should preferably not exceed one page in length in total
If you have previously published part of the work, you should say so in a few words at the end of the introduction
Response 3: We have shortened some redundant information and briefly described the current problems and proposed solutions. After deletion and modification, we reduced the length of the introduction to one page. We have adopted your opinion and added the results published previously at the end of the introduction. (line 75)
Point 4: Materials and Methods
Methods are usually been described in the order in which they were used
Unless a previously method is generally known, the reader will appreciate being told in essential features
In the statistical analysis you must include the number of the SPSS version used and the city of publication.
Response 4: We redescribed the methods in the order they were used. All the experimental methods are examined carefully and the vague and oversimplified descriptions are revised and supplemented (line 91-98). The version of SPSS and the publishing city have been supplemented (line 193-194).
Point 5: Results
In the legend of Figure 1 you must include the meaning of the diverse asterisks numbers, on the variable comparations
In the legend of Figure 5, section A, you have not shown if there are statistical differences between groups
In the same figure 5, in section C, you must explain the meaning of the acronymus LEfSe, because I didn´t found it
Response 5: We have added the meanings of the diverse asterisk numbers on the variable comparations in the legend of Figure 1 (line 200-201). Figure 5 was redrawn and statistical differences between groups were added in Part A of the legend. The meaning of the acronymus LEfSe has been explained.(line 285)
Point 6: Discussion
The first part is dedicated to explain the main message answering the question posed in the introduction and the main supporting evidences
It is convenient to insist in a critical assessment about the shortcomings in the study design, limitation in methods, flaws in analysis or validity of assumptions
Finally, is convenient to make a comparison with other studies, where inconsistencies are discussed
Response 6: We greatly appreciate your valuable suggestions. We added a paragraph at the beginning of the discussion section to answer the questions raised in the introduction and the main supporting evidence(303-308). According to your suggestion, the design flaws(373-377), methodological limitations(326-328), analytical flaws(line 457-458), and validity of the hypothesis(line 510-511) of this study was discussed.
Point 7: Conclusions
Are correct and open new proposals for future research on this field
Response 7: We have revised the conclusion and added open new proposals for future research on this field(519-525).
Point 8: References
Are well selected and appropriated
Response 7: All the references were carefully examined and the inappropriate parts were deleted.

Round 2
Reviewer 1 Report
Dear authors,
After all these corrections, I congratulate you for taking the time to improve the quality of your work and manuscript.
"How microwave treatment of gluten affects its toxicity for celiac patients? A study on the effect of microwaves on the structure, conformation, functionality and immunogenicity of gluten" Hamida Mahroug et al. This work deserves to be cited.
Best regards